# A Narrative Mini Review on Current Status of Hypoallergenic Wheat Development for IgE-Mediated Wheat Allergy, Wheat-Dependent Exercise-Induced Anaphylaxis

**DOI:** 10.3390/foods12050954

**Published:** 2023-02-23

**Authors:** Eishin Morita, Hiroaki Matsuo, Kunie Kohno, Tomoharu Yokooji, Hiroyuki Yano, Takashi Endo

**Affiliations:** 1Department of Dermatology, Shimane University Faculty of Medicine, Izumo 693-8501, Japan; 2Department of Pharmaceutical Services, Graduate School of Biomedical and Health Sciences, Hiroshima University, Hiroshima 734-8551, Japan; 3Department of Clinical Trial Management, Clinical Research Center, Shimane University Hospital, Izumo 693-8501, Japan; 4Department of Frontier Science for Pharmacotherapy, Graduate School of Biomedical and Health Sciences, Hiroshima University, Hiroshima 734-8553, Japan; 5National Food Research Institute, National Agriculture and Food Research Organization, Tsukuba 305-8642, Japan; 6Kyoto University, Kyoto 606-8501, Japan

**Keywords:** deamidation, enzymic degradation, hypoallergenic wheat, ω5-gliadin, thioredoxin, wheat-dependent exercise-induced anaphylaxis

## Abstract

Immunoglobulin E (IgE)-mediated food allergies to wheat that develop after school age typically shows a type of wheat-dependent exercise-induced anaphylaxis (WDEIA). At present, avoidance of wheat products or postprandial rest after ingesting wheat is recommended for patients with WDEIA, depending on the severity of the allergy symptoms. ω5-Gliadin has been identified as the major allergen in WDEIA. In addition, α/β-, γ-, and ω1,2-gliadins, high and low molecular weight-glutenins, and a few water-soluble wheat proteins have been identified as IgE-binding allergens in a small proportion of patients with IgE-mediated wheat allergies. A variety of approaches have been manufactured to develop hypoallergenic wheat products that can be consumed by patients with IgE-mediated wheat allergies. In order to analyze such approaches, and to contribute to the further improvement, this study outlined the current status of these hypoallergenic wheat productions, including wheat lines with a reduced allergenicity that are mostly constructed for the patients sensitized to ω5-gliadin, hypoallergenic wheat by enzymic degradation/ion exchanger deamidation, and hypoallergenic wheat by thioredoxin treatment. The wheat products obtained by these approaches significantly reduced the reactivity of Serum IgE in wheat-allergic patients. However, either these were not effective on some populations of the patients, or low-level IgE-reactivity to some allergens of the products was observed in the patients. These results highlight some of the difficulties faced in creating hypoallergenic wheat products or hypoallergenic wheat lines through either traditional breeding or biotechnology approaches in developing hypoallergenic wheat completely safe for all the patients allergic to wheat.

## 1. Introduction

Considering the immune response to wheat, wheat allergies can be divided into immunoglobulin E (IgE)- and non-IgE-mediated reactions [1]. The pathophysiology of the IgE-mediated wheat allergy involves the production of specific IgE to wheat allergens and the subsequent activation of mast cells and basophils by cross-linking IgE with wheat allergens. In contrast, a non-IgE-mediated reaction is an esophageal or gastrointestinal inflammation caused by T lymphocytes and eosinophils activated in response to wheat allergens. However, the precise mechanism underlying the reaction is unclear. The IgE-mediated wheat allergy is caused either by ingestion of wheat (food allergy) or inhalation of wheat (airway allergy called Baker’s asthma). The clinical features of IgE-mediated food allergies due to wheat are characterized by the age of the patient during the onset of the allergy. In infancy, it develops mainly in association with atopic dermatitis (AD), whereas after school age, it can affect individuals without any history of AD [2].

The IgE-mediated food allergies due to wheat that appear during childhood generally develop resistance at a high rate [2]. According to a study at the Johns Hopkins Pediatric Allergy Clinic, 65% of the children become resistant to wheat allergies up to the age of 12 years [3]. In contrast, IgE-mediated wheat allergies that develop after school age typically shows a type of wheat-dependent exercise-induced anaphylaxis (WDEIA), a life-threatening type of IgE-mediated wheat allergy. WDEIA causes allergic symptoms through a combination of secondary factors, such as exercise, drugs, alcohol, and stress, in addition to wheat ingestion, and often causes anaphylactic shock, whereas patients with WDEIA usually ingest wheat products safely without such co-factors [4]. The roles of these co-factors have been hypothesized to increase gastrointestinal permeability, tissue transglutaminase activation in the gut mucosa, blood flow redistribution, plasma osmolarity causing basophil histamine release, and acidosis causing mast cell degranulation in the tissues [5]. In order to clarify the pathophysiology of WDEIA, wheat allergens involved in the sensitization of WDEIA have been intensively investigated. The biochemical analyses indicated that ω5-gliadin is the major allergen among wheat gluten proteins, and an allergen-specific IgE test using recombinant ω5-gliadin identified the patients with WDEIA with a high sensitivity and specificity [6]. A natural history observational study reported that sensitization to wheat allergens in WDEIA continues for a long period once it develops [7]. Recent studies by Gupta et al. estimated that the prevalence of IgE-mediated wheat allergies was 0.8% in adults in the US [8]. According to an epidemiological study of local adult residents in Shimane Prefecture, Japan, the prevalence of this disease was 0.21% [9].

Allergen immunotherapy has been used for the IgE-mediated food allergies and can increase the threshold of reactivity to a variety of foods [10]. Wheat oral immunotherapy can be an effective and safe treatment modality for the children with a history of wheat anaphylaxis [11]. However, no immunotherapy or prophylaxis has been established for WDEIA. Avoiding wheat or postprandial rest is currently recommended, depending on the severity of the allergy symptoms, for patients with WDEIA [12]. Wheat is used in a variety of market foods due to its high processing characteristics; therefore, there is still a risk of life-threatening anaphylaxis due to accidental exposure, and the burden on the patients and their families is high. A variety of approaches have been made to develop wheat products that can be consumed by patients with IgE-mediated wheat allergies. However, hypoallergenic wheat products to meet the patient’s needs have not been supplied yet. In order to analyze such approaches, and to contribute to the further improvement, this study outlined the current status of the hypoallergenic wheat developed for IgE-mediated wheat allergies typically showing a type of WDEIA.

## 2. WDEIA and the Responsible Allergens

According to their chemical properties, wheat proteins are classified as water-soluble albumin, salt-soluble globulins, aqueous alcohol-soluble gliadins, and diluted acid- or alkali-soluble glutenins [13,14]. Gliadins are further classified as α/β-, γ-, and ω-gliadins based on electrophoretic mobility, whereas glutenins are classified as high-molecular weight (HMW) (67,000–88,000 Da) and low-molecular weight (LMW) (32,000–35,000 Da) subunits. Further, based on the N-terminal amino acid sequences, ω-gliadins are classified as ω1,2-gliadins (sequences beginning with ARE/KELQS) and ω5-gliadin (sequences beginning with SRLL).

To date, a variety of allergens and their IgE-binding epitopes have been identified according to the type of wheat allergy [15]. Table 1 contains a list of the wheat allergens relevant to WDEIA. ω5-Gliadin and HMW-glutenin have been identified as major allergens associated with WDEIA [16,17,18,19,20,21]. cDNA cloning of ω5-gliadin revealed that the major IgE-binding sites (epitopes) of ω5-gliadin were QQX_1_PX_2_QQ (X_1_ is L, F, S or I and X_2_ is Q, E or G) [19,20,21,22]. The major IgE epitopes of HMW-glutenin are QQPGQ, QQPGQGQQ, and QQSGQGQ [19,20]. The evaluation of sensitization rates using ω5-gliadin- and HMW-glutenin-specific IgE tests showed that more than 80% of patients with WDEIA were sensitized with ω5-gliadin, and approximately 10% of patients were sensitized with HMW-glutenin [23]. Since the sensitization rate to ω5-gliadin accounts for more than 90% of adult patients with WDEIA, WDEIA has been referred to as the “ω5-gliadin allergy” [24]. α/β-Gliadin, γ-gliadin, and ω1,2-gliadin have also been identified as allergens in patients with wheat allergy, including children with AD [15,22]. The LMW glutenin subunit was reported to be reactive with IgE in the adult patients with WDEIA, carrying specific IgE epitopes independent of ω5-gliadin epitopes [25].

Owing to their close phylogenetic relationship, secalins of rye and hordeins of barley share closely related amino acid sequences with wheat gluten proteins [14]. HMW-glutenins, HMW-secalins, and D-hordeins are classified into the HMW group, ω1,2-gliadin, ω5-gliadin, ω-secalins, and C-hordeins are classified into the medium MW group, and LMW-glutenins, α-gliadins, γ-gliadins, γ-75k-secalins, γ-40k-secalins, B-hordeins, and γ-hordeins are classified into the LMW groups [28]. Using an enzyme-linked immunosorbent assay (ELISA) with monoclonal antibodies against gluten, cross-reactivity among gluten proteins, secalins, and hordeins was established [28]. The cross-reactivity of γ-70 (γ-75k) and γ-35 (γ-40k) secalins in rye and γ-3 hordein in barley with ω5-gliadin was shown by ELISA using sera from patients with WDEIA and a skin prick test in these patients [29], and the existence of IgE binding epitopes of ω5-gliadin were shown in the γ-75k and γ-35 (γ-40k) secalins and γ-3 hordein [30].

An outbreak of wheat allergy (mostly WDEIA) caused by hydrolyzed wheat protein (HWP) occurred in Japan from 2008 to 2010 [31,32,33]. This was caused by cutaneous sensitization during the use of soap bars containing HWP. IgE against HWP cross-reacts with orally ingested wheat products. These hydrolyzed wheat allergies differ from conventional WDEIA with respect to the negative or low levels of ω5-gliadin-specific IgE. Yokooji et al. used recombinant allergens of wheat constituent proteins to study wheat proteins recognized by IgE in the serum of patients with hydrolyzed wheat allergies. They found that γ-gliadin was the major allergen and its major epitope was QPQQPFPQ [26]. This epitope is consistent with the IgE epitope QPEEPFPE of the hydrolysates of γ-gliadin and ω2-gliadin identified in patients with hydrolyzed wheat allergies in Europe [34].

Recently, wheat peroxidase-1 and β-glucosidase have been identified as specific IgE-binding allergens that cross-react with grass pollen allergens in the patients with WDEIA who developed a grass pollen allergy [27,35].

## 3. Approaches to Develop Hypoallergnic Wheat

In order to develop wheat products that can be consumed by patients with IgE-mediated wheat allergies, especially WDEIA, a variety of approaches were developed to remove the major wheat allergens described above. These are wheat lines with a reduced allergenicity that are mostly constructed for the patients sensitized to ω5-gliadin, hypoallergenic wheat by enzymic degradation/ion exchanger deamidation, and hypoallergenic wheat by thioredoxin treatment. Wheat lines with a reduced allergenicity were established using either natural occurring wheat deletion lines of allergen genomes or transgenic wheat lines with RNA interference technique. Hypoallergenic wheat were produced by either epitope degradation with specific enzymes after identifying their epitopes using serum IgE of the patients with wheat allergy complicated with AD, or deaminating amino groups of glutamine and/or asparagine within the epitopes by treating with cation exchange resin. A subsequent attempt was made to dissociate disulfide bonds in gluten proteins, such as gliadin and glutenin, to reduce allergenicity using reducing agent thioredoxin. These methods to develop hypoallergenic wheat, and their outcomes, are summarized in Table 2.

### 3.1. Wheat Lines with a Reduced Allergenicity

The major allergen in WDEIA is ω5-gliadin, which accounts for only a minor proportion of gluten. Among several ω-gliadins, ω1,2-gliadins are encoded by *Gli-A1* and *Gli-D1*loci on the short arms of the Group 1 chromosome, whereas ω5-gliadin is encoded on the *Gli-B1* locus on chromosome 1B [65,66]. Denery-Papini, et al. investigated 13 wheat cultivars with genetic variability at the *Gli-B1* locus for reactivity to rabbit ω5-gliadin-specific antiserum and IgE from 10 patients, including those with WDEIA. They found that 1BL/1RS translocated wheat, in which a part of the short arm of the 1B chromosome was replaced with a portion of the short arm of the 1R chromosome of rye, lost reactivity to ω5-gliadin when tested using rabbit ω5-gliadin-specific antiserum [36]. In addition, the reactivity of Serum IgE from the patients against its gliadin preparation was mostly lost, except for one patient with anaphylaxis, who had IgE reacting with a 44 kDa band corresponding to 1RS-encoded ω-secalin, indicating that this line may be beneficial for patients with WDEIA. Gabler et al. compared the allergenicity of gluten prepared from another wheat/rye translocation line, Pamier (ω5-gliadin content; 2.40 mg/g protein), and gluten prepared from a conventional wheat line (22.3 mg/g protein) using the CD63-monitored basophil activation test in 12 patients with WDEIA. However, no significant difference was observed in the basophil activation between these two gluten preparations [37,38]. These findings suggest that non-gluten proteins, possibly ω-secalin derived from 1RS, are relevant because of their cross-reactivity with ω5-gliadin. Wheat lines carrying this translocation generally had poor bread-making qualities [67].

Lombardo et al. examined the allergenicity of *Triticum monococcus*, an earliest cultivated A genome diploid einkorn, in 14 patients with WDEIA sensitized with ω5-gliadin. A skin prick test using soluble and insoluble extracts failed to induce reactivity in almost all the patients tested. IgE-immunoblotting showed an absence of ω5-gliadin in the proteins of *Triticum monococcus*, although a variety of reactions were observed with limited cross-reactivity to ω5-gliadin [39]. As some einkorn accessions possess good bread-making characteristics, *Triticum monococcus* might be a potential candidate for the production of hypoallergenic bakery products in patients sensitized to ω5-gliadin [68].

Using a traditional breeding method, Waga, et al. established a winter wheat line possessing hybrid genotypes lacking all ω-gliadin coding loci (*Gli A1, Gli B1 and Gli D1*) and investigated the reactivity of Serum IgE in several patients with wheat allergies (including asthma, rhinitis, urticaria/angioedema, anaphylaxis, and AD). Although IgE binding to 60 kDa ω5-gliadin was lost in the hybrid line only in one patient with possible WDEIA, the hybrid line was immune-reactive against patients’ IgE with many gluten proteins, including α/β-, γ-gliadins, LMW-glutenin, and non-gluten proteins [40,41].

Kohno et al. found that one of the deletion lines of Chinese Spring wheat (1BS-18), in which the end of the short arm of chromosome 1B was deleted, lacked the *Gli B1* locus. They confirmed the lack of ω5-gliadin protein in this line using immunoblotting with rabbit polyclonal antibodies against the ω5-gliadin epitope peptide, as well as a reversed phase-high performance liquid chromatography. They also confirmed the low allergenicity of 1BS-18 in a guinea pig challenge model [42]. Furthermore, this line (an experimental line unsuitable for practical use) was backcrossed with the Japanese practical wheat cultivar Hokushin, and the ω5-gliadin-deficient wheat line (1BS-18 Hokushin) was established.

Yokooji et al. found that the ω5-gliadin content of 1BS-18 Hokushin was 1.21 mg/g gluten, and this is much lower than that of euploid Hokushin (5.17 mg/g gluten) and another Japanese wheat cultivar Norin 61 (5.65 mg/g gluten); ELISA with rabbit polyclonal antibodies specifically recognizing the IgE-binding epitope sequences (KQQSPEQQQFPQQQIPQQQ) of ω5-gliadin was used for the assessment. They speculated that the slight detection of ω5-gliadin in the 1BS-Hokushin was due to cross-reactivity with gliadin components other than ω5-gliadin [43]. Yamada et al. evaluated the allergenicity of 1BS-18 Hokushin using a rat wheat-anaphylaxis model and found that gluten proteins of 1BS-18 Hokushin elicited no allergic reaction in ω5-gliadin-sensitized rats and had less sensitization ability to ω5-gliadin than those of euploid Hokushin wheat [44]. In addition, they found that early consecutive ingestion of 1BS-18 Hokushin prevents subcutaneous immunization against ω5-gliadin protein using the rat wheat-anaphylaxis model, suggesting that 1BS-18 Hokushin induces oral tolerance to wheat allergens [45].

The hypoallergenicity of 1BS-18 Hokushin and another wheat line, 1BS-18 Minaminokaori, was investigated using immunoblotting, and the lack of ω5-gliadin was determined using Serum IgE, which was obtained from patients with WDEIA as well as rabbit polyclonal antibodies against ω5-gliadin epitope peptide. The evaluation for allergenicity showed faint or no reaction bands corresponding to ω5-gliadin (Figure 1). Although the safety of 1BS-18 wheat products for the patients with WDEIA should be confirmed by clinical studies, the introduction of wheat lacking ω5-gliadin into wheat products could reduce the chance of exposure of consumers to ω5-gliadin and, hence, the population of patients with WDEIA.

Lee et al. investigated the allergenicity of wheat mutant line DH20, with a defect in the chromosome B *Glu-B3* and *Gli-B1* loci having selective deletions in ω5-gliadin, as well as some LMW glutenins and γ-gliadins using IgE-immunoblotting with sera from patients with WDEIA, and found that the gliadin and glutenin fractions of DH2 had less binding of IgE from the patients compared with that of the wild type wheat line [46,47]. In addition, the ELISA inhibition assay showed that 50% inhibitory concentrations of DH2 fractions against gliadin- or glutenin-IgE reactivity were approximately 4-fold higher than those of wild-type wheat line.

Notably, two-dimensional immunoblot analysis revealed the existence of several minor but highly immunogenic ω5-gliadin proteins in a wheat mutant line, DH20, with a deletion of 5.8 Mb, including the *Gli-B1* locus of Chromosome 1B [48]. These proteins were found to be encoded in the ω5-gliadin gene of Chromosome 1D and to have a high homology, including repetitive IgE binding epitopes to the ω5-gliadin of the *Gli-B1* locus, although they have a TRQ N-terminal amino acid sequence and altered C-terminal amino acid sequence compared with the *Gli-B1* ω5-gliadin [49]. It is not known whether other hexaploid wheat cultivars contain such active ω5-gliadin genes on the 1D chromosome. These findings indicate the importance of a detailed understanding of the gluten protein genes in individual cultivars for exploring hypoallergenic wheat.

Altenbach et al. showed that the ω5-gliadin protein was either undetectable or depleted in two transgenic wheat lines with reduced levels of ω5-gliadin using the RNA interference technique [50,51]. IgE-immunoblotting using the products of these transgenic wheat lines revealed that the reactivity to ω5-gliadin of the serum IgE was greatly reduced in seven of 11 patients with WDEIA [52]. These findings indicate that the response of patients with WDEIA to ω5-gliadin can be effectively eliminated by changing only the protein encoded by the *Gli-B1* locus. In the short-to-medium term, these wheat products are expected to be costly because of the high cost of the cultivar development using the transgenic processing. Therefore, it may be more appropriate to supply these products to ω5-gliadin-sensitized patients with WDEIA, who are restricted from using wheat products.

### 3.2. Hypoallergenic wheat by Enzymic Degradation/Ion Exchanger Deamidation

Based on the results of IgE-binding epitope analysis of wheat allergens, hypoallergenic wheat flour was subjected to enzymatic modification or deamidation. Tanabe et al. identified IgE epitopes of gluten protein allergens using Serum IgE from patients with AD. They first identified a 30-mer peptide corresponding to a part of the amino acid sequence of LMW glutenin, and then determined that QQQPP is the smallest unit recognized by the patient’s Serum IgE. Substitution experiments with glycine revealed that the amino acids essential for IgE binding are the first Q, and the fourth and the fifth P [69]. Pastorello et al. also reported that LMW glutenin is one of the causative allergens of wheat allergy in children with AD [70]. Watanabe et al. searched for various proteases to cleave the IgE epitope QQQPP and devised a two-step method for producing a hypoallergenic wheat product, cupcake, using cellulase and actinase [53]. The hypoallergenic wheat flour consists mainly of oligopeptides and amino acids, and its average molecular weight was lower than 1000 [54]. The safety of the hypoallergenic wheat flour was evaluated using a DNA microarray in rats, and no groups of genes known to be involved in the carcinogenesis or oxidative stress were affected [55]. When the cupcakes prepared by this method were ingested by 15 children with AD, a systemic urticaria was induced in two children, whereas 13 children were able to ingest cupcakes without adverse effects [56,57]. Furthermore, an open study of oral immunotherapy with continuous cupcake intake showed that most of the 20 children with a history of wheat allergy became able to consume normal wheat products, suggesting that the continuous intake of hypoallergenic wheat products could induce oral immunotolerance [57]. Future studies should examine whether the hypoallergenic wheat products produced by the two-step method using cellulase and actinase are effective in inducing immunotolerance in the children with wheat allergies developed in association with AD.

Recently, it has been clarified that the food allergies observed in children with AD are not caused by food allergens ingested orally, but by the cutaneous sensitization caused by a trace amount of food allergens that invade the lesions of AD [71]. Since a variety of food allergens exist in the environment, the sensitizing food allergens in children with AD are very diverse. The use of emollients in infancy does not always prevent the development of AD and food allergies [72]. The early initiation of food intake in infants is also not always effective in preventing food allergies [71]. A tight control of dermatitis using topical corticosteroids at the early stage of AD decreases the risk of development of food allergies [73,74].

Some reports have shown that the deamidation of wheat gliadin decreases allergenicity. Kumagai et al. performed the deamidation of gliadin using a cation exchange resin without affecting peptide-bond and investigated its allergenicity using a rat model as well as patients’ Serum IgE. Compared with undeamidated gliadin, the deamidated gliadin showed lower reactivity against Serum IgE obtained from patients with high levels of wheat-specific IgE, and induced lower levels of gliadin-specific IgE in the rat allergy model of oral administration [58]. Abe et al. examined the allergenicity of deamidated gliadin in a mouse model of wheat-gliadin allergy. An oral administration of the deamidated gliadin suppressed intestinal permeability, serum allergen levels, serum allergen-specific IgE levels, mast cell-surface expression of FcεRI, and serum and intestinal histamine levels [59]. On the other hand, Abe et al. also clarified that deamidated and hydrolyzed gliadin induced severe allergic reactions, while deamidated-only and hydrolyzed-only gliadin showed almost no allergic response in the transdermal administration model of mice [75]. These findings are compatible with the outbreak of hydrolyzed wheat protein allergies described above [26,31,32,33,34].

### 3.3. Production of Hypoallergenic Wheat by Thioredoxin Treatment

Disulfide bonds usually provide a digestion-resistant feature and increase the allergenicity of food proteins [76]. Using a canine allergy model, Buchanan et al. showed that thioredoxin, reduced by NADPH via NADP-thioredoxin reductase, mitigated the allergenicity of wheat proteins, particularly gliadin and glutenin, by reducing their disulfide bonds. This decrease occurred along with an increased susceptibility to proteolysis, heat denaturation, and altered biochemical activity [60]. Yano et al. explored a technique to identify the target proteins of thioredoxin using electrophoresis after labelling with a fluorescent probe, providing information to produce practical hypoallergenic wheat products by thioredoxin treatment [61,62]. Waga et al. analyzed 10 winter wheat genotypes treated with thioredoxin using ELISA with Serum IgE obtained from patients and found that the reduction by thioredoxin strongly decreased gliadin immunoreactivity but did not significantly affect dough rheological properties [63]. Matsumoto et al. investigated effects of thioredoxin on the allergenicity of salt-soluble wheat proteins in six patients with IgE-mediated wheat allergy and found that the thioredoxin-treated wheat proteins mitigated both the reaction of the skin prick test and the binding to serum IgE by inhibition assay using fluorescence enzyme immunoassay using ImmunoCAP (CAP-FEIA) [64]. An overexpression of thioredoxin in the wheat endosperm was found to increase the solubility and decrease the allergenicity of gliadins in a canine allergy model, indicating that the high expression in seeds by gene recombination reduces their allergenicity [77].

Since ω-gliadins contain no cysteine residues, they do not participate in the formation of the disulfide bridges that stabilize the gluten protein structure. However, the ω-gliadins interact with other proteins via weak, low-energetic hydrogen bonds. Stawoska et al. suggested that the elimination of ω-fractions from the gliadin complex causes minor modifications to the secondary structures of the remaining gliadin proteins, facilitating the interaction of IgE epitopes with IgE antibodies [78]. NADPH and reductase, which activate thioredoxin, are required to act on thioredoxin. It has also been reported that thioredoxin itself may be allergenic [79,80,81]. At present, consumers tend not to like the use of genetic modification or additives in food [82]. The use of thioredoxin in food products should be carried out carefully while monitoring costs and consumer awareness.

## 4. Conclusions

The limitations of this study are that we have not presented details of the methods that have been used for developing the hypoallergenic wheat, and that the references written in English have been analyzed while the five references written in non-English languages have been omitted in the analysis among 61 references extracted using PubMed with the keyword “hypoallergenic wheat.”

Several approaches have been used to establish hypoallergenic wheat products that can be consumed by patients with IgE-mediated wheat allergies. These approaches are divided globally into two methods. One is the alteration of allergen epitopes by enzymatic degradation, deamidation, or the addition of reducing agents, and the other is the production of wheat lines that do not contain major allergen epitopes by traditional breeding or biotechnology. In particular, hypoallergenic wheat lines, which lack ω5-gliadin with transgenic or natural breeding techniques, significantly reduced the reactivity of Serum IgE in wheat-allergic patients while retaining the characteristics of flour, such as bread-making properties. However, low-level reactivity to wheat allergens in the wheat-allergic patients has also been observed in these wheat lines. This may be due to cross-reactivity between the epitopes of ω5-gliadin and similar amino acid sequences of other gluten proteins. It is also possible that the wheat-allergic patients have Serum IgE that reacts primarily with α/β-, γ-, and ω1,2-gliadins, or HMW- and LMW-glutenins. Given the complexity of the immune response of patients with WDEIA and the severity of the allergic reaction, it is not feasible for the patients with WDEIA to consume flour from wheat lines with a reduced allergenicity without a thorough analysis of the IgE reactivity of each patient’s serum. Better diagnostic methods should be developed to define sensitization conditions precisely in conjunction with hypoallergenic wheat products. These results highlight some of the difficulties faced in creating new hypoallergenic wheat lines through either traditional breeding or biotechnology approaches in developing hypoallergenic wheat for patients allergic to wheat. Nevertheless, the provision of new wheat lines with a reduced allergenicity in general populations may reduce the number of subjects sensitized to wheat proteins in the future. Subjects possesing HLA-DPB1^∗^02:01:02, a susceptibility gene for WDEIA, could especially avoid a sensitization to ω5-gliadin if they would consume these hypoallergenic wheat products in their daily meals [83]. Hypoallergenic wheat could also be applied in new immunotherapy protocols aimed at desensitizing patients to specific wheat allergens.

## Figures and Tables

**Figure 1 foods-12-00954-f001:**
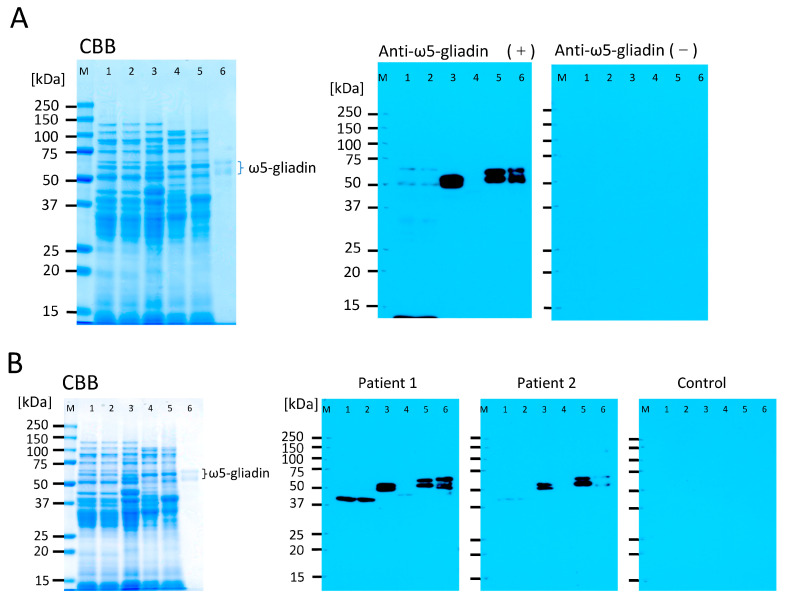
Immunoblotting to detect immunoreactive ω5-gliadin in ω5-gliadin-deficient wheat lines (1BS-18 Hokushin and 1BS-18 Minaminokaori). Immuoblotting was performed for wheat flour samples (10 μg/lane) and purified ω5-gliadin (0.5 μg/lane) with polyclonal rabbit anti-ω5-gliadin IgG antibody [42] (**A**), and sera obtained from two patients with WDEIA (**B**). Lane M; molecular weight marker, lane 1; 1BS-18 minaminokaori obtained from field 1, lane 2; 1BS-18 minaminokaori obtained from field 2, lane 3; euploid Minaminokaori, lane 4; 1BS-18 Hokushin, lane 5; euploid Hokushin, lane 6; purified ω5-gliadin. Patient 1; serum ω5-gliadin-specific IgE 16.5 UA/mL, Patient 1; serum ω5-gliadin-specific IgE 25.5 UA/mL, Contorl; serum ω5-gliadin-specific IgE <0.35 UA/mL, as determined by ImmunoCAP (rTri a 19: ω5-gliadin) (ThermoFischer Diagnostics, Waltham, MA, USA).

**Table 1 foods-12-00954-t001:** What allergens associated with WDEIA.

Allergens	Common Name	MW (kDa)	Route of Allergen Exposure	Ref. No.
Tri a 19	ω5-gliadin	65	Food (Ingestion)	[15,16,17,18,19,20,21]
Tri a 20	γ-gliadin	35–38	Food (Ingestion), HWP (cutaneous)	[15,22,26]
Tri a 21	α/β-gliadin	28–35	Food (Ingestion)	[15,22]
Tri a 26	HMW-glutenin	67–88	Food (Ingestion)	[15,20]
Tri a 36	LMW-glutenin	32–35	Food (Ingestion)	[15,25]
—	ω1,2-gliadin	40	Food (Ingestion), HWP (cutaneous)	[15,22,26]
—	Peroxidase-1	36	Cross-reactivity to grass pollen allergens	[27]
—	β-glucosidase	60	Cross-reactivity to grass pollen allergens	[27]

WDEIA: wheat-dependent exercise-induced anaphylaxis, MW: molecular weight, HWP: hydrolyzed wheat protein, HMW: high molecular weight, LMW: low molecular weight, —: not listed in the allergen nomenclature by WHO/IUIS.

**Table 2 foods-12-00954-t002:** Summary of the methods to develop hypoallergenic wheat and their outcomes.

Methods	Major Outcomes and Limitations	Ref. No.
**Wheat lines with a reduced allergenicity**		
1BL/1RS translocation (Clement)	Significant reduction of ω5-gliadin protein was presented using SDS-PAGE and immunoblotting with rabbit polyclonal antibody. Low allergenicity was presented with IgE-immunoblotting using small numbers of WDEIA patients’ sera. No oral challenge test was preformed in human.	[36]
1B/1R translocation (Pamier)	Significant reduction of ω5-gliadin protein was determined by RP-HPLC. No significant difference in basophil activation compared with conventional wheat line in WDEIA patients.	[37,38]
A genome diploid einkorn lacking B chromosomes	Low allergenicity was presented in negative skin prick test and lack of IgE-immunoreactivity to ω5-gliadin in almost all WDEIA patients sensitized with ω5-gliadin. No oral challenge test was performed in human.	[39]
Wheat line lacking all ω-gliadin encoding loci *Gli A1*, *Gli B1* and *Gli D1* established using traditional breeding (3xN)	Lack of ω5-gliadin was determined using SDS-PAGE and RP-HPLC. Low reactivity to ω5-gliadin was presented using ELISA with several WDEIA patients’ sera. Significant IgE reactivity was presented to other gluten proteins. No oral challenge test was performed in human.	[40,41]
Aneuploid wheat line lacking *Gli-B1* locus (1BS-18)	Lack of ω5-gliadin protein was determined using RP-HPLC and immunoblotting with rabbit polyclonal antibody. Low allergenicity was presented using guinea pig and rat wheat challenge models. Oral tolerance to wheat was determined using rat models. No oral challenge test was performed in human.	[42,43,44,45]
Wheat line lacking *Gli B1* and *Glu B3* loci established using double-haploid breeding (DH20)	Low allergenicity was determined using IgE-immunoblotting with 14 WDEIA patients’ sera, immunoblot inhibition assay and ELISA inhibition assay. No oral challenge test was performed in human.	[46,47,48,49]
**Genetic engineering**		
Transgenic wheat lines with RNA interference for ω5-gliadin (Butte 86)	ω5-Gliadin protein was either undetectable or depleted in SDS-PAGE. Greatly reduced IgE-immunoreactivity to ω5-gliadin was presented in most of 11 WDEIA patients using IgE-immunoblotting. Small amount of ω5-gliadin protein derived from 1D chromosome remaind in the line. No oral challenge test was performed in human.	[50,51,52]
**Enzymic degradation**		
Two-step method to produce hypoallergenic wheat using cellulase and actinase	Low allergenicity and immunotolerance effect were confirmed by clinical open studies. Thirteen of 15 children with AD ingested cupcake made of the wheat without allergy symptom. Most children with wheat allergy associated with AD became immunotolerance to normal wheat product after continuous intake of the cupcake.	[53,54,55,56,57]
**Ion exchanger deamidation**		
Deamidation of gliadin using a cation exchange resin	Low allergenicity was presented using IgE-immunoblotting with sera obtained from patients with High Serum IgE, and a mouse model of wheat-gliadin allergy. No clinical study was performed in human.	[58,59]
**Thioredoxin treatment**		
Reduced thioredoxin treatment of wheat allergens	Low allergenicity was presented using ELISA with Serum IgE obtained from patients with wheat allergy, skin prick test and CAP-FEIA inhibition assay. No oral challenge test was performed in human.	[60,61,62,63,64]

SDS-PAGE: sodium dodecyl sulfate-polyacrylamide gel electrophoresis, WDEIA: wheat-dependent exercise-induced anaphylaxis, RP-HPLC: reversed phase-high performance liquid chromatography, ELISA: enzyme-linked immunosorbent assay, AD: atopic dermatitis, CAP-FEIA: fluorescence enzyme immunoassay using ImmunoCAP.

## Data Availability

The data that support the findings of this study are available from the corresponding author upon request.

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
