# Peer review of "A Narrative Mini Review on Current Status of Hypoallergenic Wheat Development for IgE-Mediated Wheat Allergy, Wheat-Dependent Exercise-Induced Anaphylaxis"

_foods, 2023, doi:10.3390/foods12050954_

Round 1

Reviewer 1 Report

The article by Morita et on the "Current Status of hypoallergenic wheat Development ..." is an excellent review article on current advances directed at wheat dependent exercise-induced anaphylaxis patients.  The article describes previous approaches of two types either alteration of epitopes or genetic manipulation of the crop.  The advantages and drawbacks of each are adequately described.  The article covers the current state of the art.

Very minor typos noted:

line 63 WEDIA should be WDEIA

line 312 suspect that theological should be rheological.  Otherwise this makes a very humorous sentence.

Author Response

Reply to the comments

Comments from Reviewer 1

The article by Morita et on the "Current Status of hypoallergenic wheat Development ..." is an excellent review article on current advances directed at wheat dependent exercise-induced anaphylaxis patients.  The article describes previous approaches of two types either alteration of epitopes or genetic manipulation of the crop.  The advantages and drawbacks of each are adequately described.  The article covers the current state of the art.

Reply to the comment. Thank you for your favorable comment.

Very minor typos noted:

1: line 63 WEDIA should be WDEIA

Reply to minor comment 1. I corrected WEDIA to WDEIA in line 108 in the revised manuscript. Thank you for your kind indication.

2: line 312 suspect that theological should be rheological.  Otherwise this makes a very humorous sentence.

Reply to minor comment 2. This is my mistake. I have corrected to rheological in line 357 in the revised manuscript. Thank you again.

Reviewer 2 Report

The authors have chosen an important topic to write a review article on. However, the article, in its current format, looks like a short communication. It does not give the ‘current status of the chosen topic. The manuscript provides succinct information. The authors should explain the main techniques they have mentioned. For developing a hypoallergenic wheat type. The authors have only summarized only the main finding of the previous works which is not enough for a review article. My specific comments are below.    

Major Comments

Comment#1: The authors have only written a ‘collection of previous works’ conducted.  The authors must provide their own insight into those works. They must discuss the impact of the previous studies rather than only the main findings, future directions, etc.

Comment#2: Sections 2-5 only discuss previous works. The authors may summarize these works in a table and discuss the impact of the work (developing low allergic wheat). Or please summarize the work (do not only write the findings). For example, please discuss succinctly how the authors conducted the work. (Line no. 260).

Comment#3: Some previous studies that the authors have discussed are outdated (the year 1976, 2000). I am not sure how such studies influence the current advanced study approaches. However, if those works have been the landmark studies, the authors must present them in a respective way to justify their impact. For example, please write ‘in a landmark work’/ ‘in the first evidence’.

Comment#4: The authors should explain how a hypoallergenic wheat variety are being developed in recent time, and their overall scientific procedure (in summary). So that a reader can get an idea about it. As a reader, I only see some ‘terms’ without any explanation.

Comment#5: There are multiple terms like deamidation, enzymatic modifications etc. The authors must describe those terms succinctly.

Comment#6: The authors have discussed two methods to develop hypoallergenic wheat. However, they summarized the first method (enzymatic modification) in some lines only. They did not discuss the second method. They also didn't discuss how a safe wheat variety can be developed by the second method.

Comment#7: Wheat breeding has been discussed as a part of previous work. The authors did not discuss the most advance methods being used in current time like RNAi technology and CRISPR gene editing method, their pros and cons, and the work done so far using these methods. If these methods are not suitable for WA they must discuss why.

Comment#8: The introduction is weak; it lacks multiple important pieces of information. Like, the definition of WEDIA, how wheat allergy takes place, pathophysiology (whatever is known so far), is there any role of HLA? how it is different from other gluten-related disorders? what is the prevalence of the WA?

Comment#9: In the justification provided in lines number 61-67, it looks like patients with WA can prevent wheat exposure with a little effort. These lines do not justify a real need for a hypoallergenic wheat variety. I suggest adding some information that shows an unmet need for WA-safe wheat.

Comment#10: The last part of the introduction is marginally effective. Considering a review article, the introduction is quite summarized. It should explain more about the study. The authors may explain the pathophysiology of WA briefly.

Comment#11: Lines no 65-67 discuss the theme of the study. The authors must give a little more information about such wheat and what efforts have been done so far and why the authors decided to write this manuscript.

Comment#12: The position of Table 1 is not correct. According to the ‘foods’ recommendations, Ideally, It should come somewhere after line no 83 (where the table has been mentioned for the very first time). The third column 'Reference' should display the name of the author and year with the citation. Also, note that ‘Table 1’ has been mentioned two times.

Comment#13: I am not sure if the author has conducted some lab work and showed the immunoblotting pictures in Figure 1 or if they have taken the gel pictures from some other work. If the authors are using gel pictures from any previously conducted study, the authors must declare this and take permission from the journal.

Comment#14: Authors must discuss the limitations of the study.

Minor comments

Comment#1: Please discuss (in summary), why/how wheat triggers an immunological response. Line no. 33

Comment#2: Why the authors used ‘were’ instead of ‘are’? If they are discussing any previous study, I think they should mention it. for example, according to so and so et al. Then the 'were' will be justified. Line no 88-90

Comment#3: Which patients? I think again this line is a part of any previous study. if they use ‘were’ it means it is a previously conducted work. I suggest the authors if they are not using any previously conducted study better write the sentence in the present tense. Line no. 106

Comment#4: If the mentioned work is a recently published study. the authors should use ‘has/have been’ instead of ‘were’. Line no 120-122

Comment#5: This heading can be more effective. Line no 123

Comment#6: I suggest shifting reference#30 to line number 117. And if lines number 117-119 are part of the same study, cite the references 30-31 together. Also please shift reference#36 from line no 127 to line no. 136.

Comment#7: Please cite the reference at the end of the corresponding lines instead of adjacent to the author’s name.

Comment#8: Lines number 44, and 162 need reference. 

Author Response

Reply to the comments

Comments from Reviewer 2

The authors have chosen an important topic to write a review article on. However, the article, in its current format, looks like a short communication. It does not give the ‘current status of the chosen topic. The manuscript provides succinct information. The authors should explain the main techniques they have mentioned. For developing a hypoallergenic wheat type. The authors have only summarized only the main finding of the previous works which is not enough for a review article. My specific comments are below.    

Reply to the comment. Thank you for your critical comment. A short communication in medical journals has word limitation, usually less than 1000 words, whereas this review has 3550 words with 78 references, which could not be a short communication. This review outlined the current status of hypoallergenic wheat development for the patients with wheat-dependent exercise-induced anaphylaxis. Then readers easily access the references of which they want to refer. Therefore, we corrected the title to “A narrative mini review on current status of hypoallergenic wheat development for IgE-mediated wheat allergy, wheat-dependent exercise-induced anaphylaxis”.

We have studied your following comments carefully and made some corrections. We hope the revised manuscript will be suitable for publication.

Major Comments

Comment#1: The authors have only written a ‘collection of previous works’ conducted.  The authors must provide their own insight into those works. They must discuss the impact of the previous studies rather than only the main findings, future directions, etc. 

Reply to the main comment #1. In case of original paper, authors should discuss intensively on the results obtained, whereas in case of review authors outline references in the special theme for the readers easily to understand the topics. Thus, we put the discussion describing the evaluation of these attempts and future directions.

Comment#2: Sections 2-5 only discuss previous works. The authors may summarize these works in a table and discuss the impact of the work (developing low allergic wheat). Or please summarize the work (do not only write the findings). For example, please discuss succinctly how the authors conducted the work. (Line no. 260).

Reply to the main comment #2. According to your comment, we added Table 2 summarizing the methods to produce hypoallergenic wheats and their major outcomes and limitations in the revised manuscript. The subheadings were rearranged accordingly.

Comment#3: Some previous studies that the authors have discussed are outdated (the year 1976, 2000). I am not sure how such studies influence the current advanced study approaches. However, if those works have been the landmark studies, the authors must present them in a respective way to justify their impact. For example, please write ‘in a landmark work’/ ‘in the first evidence’. 

Reply to the main comment #3. As you indicated, we cited some old references (Ref. 35, 36, 58-61 in the revised manuscript). Wheat has a long history to be provided as a major energy source. The ref. 35 and 36 are a landmark article showing relations between wheat proteins and wheat chromosomes. Further, Ref. 58-61 are also important studies first showing the technique for enzymic degradation to produce hypoallergenic wheat.

Comment#4: The authors should explain how a hypoallergenic wheat variety are being developed in recent time, and their overall scientific procedure (in summary). So that a reader can get an idea about it. As a reader, I only see some ‘terms’ without any explanation.

Reply to the main comment #4. Thank you for your important comment. Hypoallergenic wheat has been developed to remove major wheat allergens causing IgE-mediated allergic reaction from wheat products. According to your comment, we briefly summarized the methods in development of hypoallergenic wheat before Table 2 in the heading 3.

Comment#5: There are multiple terms like deamidation, enzymatic modifications etc. The authors must describe those terms succinctly. 

Reply to the main comment #5. We added brief explanation for the technical terms as indicated in reply to the comment #4.

Comment#6: The authors have discussed two methods to develop hypoallergenic wheat. However, they summarized the first method (enzymatic modification) in some lines only. They did not discuss the second method. They also didn't discuss how a safe wheat variety can be developed by the second method. 

Reply to the main comment #6. As you indicated, two methods (enzymatic degradation and deamidation of epitopes) are described. Some clinical studies were carried out to see the efficacy and safety using the hypoallergenic wheat produced by enzymic degradation in the references no.60, and 61 in the revised manuscript, whereas no clinical study was performed using the hypoallergenic wheat produced by deamidation. Furthermore, the deamidated wheat proteins can obtain the higher allergenicity as described in the references no. 27-31 in the revised manuscript.

Comment#7: Wheat breeding has been discussed as a part of previous work. The authors did not discuss the most advance methods being used in current time like RNAi technology and CRISPR gene editing method, their pros and cons, and the work done so far using these methods. If these methods are not suitable for WA they must discuss why.

Reply to the main comment #7. The wheat line lacking ω5-gliadin presented by reference no. 51-53 (in the revised manuscript) was established by silencing ω5-gliadin using RNA interference technique. These are not provided for commercial use at present, because this needs high cost (this was described in the manuscript) and presumably because most people abhor genetically engineered wheat. Gene editing methods are mainly used to investigate functions of wheat proteins in the laboratories, but not for producing hypoallergenic wheat so far (we don’t find such references).

Comment#8: The introduction is weak; it lacks multiple important pieces of information. Like, the definition of WEDIA, how wheat allergy takes place, pathophysiology (whatever is known so far), is there any role of HLA? how it is different from other gluten-related disorders? what is the prevalence of the WA?

Reply to the main comment #8. Thank you for your important comment. The description for WDEIA explanation might not be enough to understand its pathophysiology in the Introduction section. We added hypotheses for the pathophysiology of WDEIA in the Introduction. The role of this review is to focus overall development of hypoallergenic wheat for WDEIA. There have already been many good reviews to describe definition, diagnosis, pathophysiology of WDEIA, including mine (reference no. 4) and another review (reference no.5 in the revised manuscript) cited in this study. The relation of WDEIA to HLA is presented in our two previous articles (Fukunaga K, Chinuki Y, Hamada Y, Fukutomi Y, Sugiyama A, Kishikawa R, Fukunaga A, Oda Y, Ugajin T, Yokozeki H, Harada N, Suehiro M, Hide M, Nakagawa Y, Noguchi E, Nakamura M, Matsunaga K, Yagami A, Morita E, Mushiroda T. Genome-wide association study reveals an association between the HLA-DPB102:01:02 allele and wheat-dependent exercise-induced anaphylaxis. Am J Hum Genet. 2021; 108: 1540-1548. doi: 10.1016/j.ajhg.2021.06.017.) and (Noguchi E, Akiyama M, Yagami A, Hirota T, Okada Y, Kato Z, Kishikawa R, Fukutomi Y, Hide M, Morita E, Aihara M, Hiragun M, Chinuki Y, Okabe T, Ito A, Adachi A, Fukunaga A, Kubota Y, Aoki T, Aoki Y, Nishioka K, Adachi T, Kanazawa N, Miyazawa H, Sakai H, Kozuka T, Kitamura H, Hashizume H, Kanegane C, Masuda K, Sugiyama K, Tokuda R, Furuta J, Higashimoto I, Kato A, Seishima M, Tajiri A, Tomura A, Taniguchi H, Kojima H, Tanaka H, Sakai A, Morii W, Nakamura M, Kamatani Y, Takahashi A, Kubo M, Tamari M, Saito H, Matsunaga K. HLA-DQ and RBFOX1 as susceptibility genes for an outbreak of hydrolyzed wheat allergy. J Allergy Clin Immunol. 144(5): 1354-1363, 2019doi: 10.1016/j.jaci.2019.06.034.) We cited these (reference no. 82 in the revised manuscript) and discussed in the Conclusion.

Comment#9: In the justification provided in lines number 61-67, it looks like patients with WA can prevent wheat exposure with a little effort. These lines do not justify a real need for a hypoallergenic wheat variety. I suggest adding some information that shows an unmet need for WA-safe wheat.

Reply to the main comment #9. Thank you for your good suggestion. We added some description to the last part (line 79-82 in the revised manuscript) “However, hypoallergenic wheat products to meet the patient needs has not been supplied. In order to analyze such approaches and to contribute to the further improvement, this study outlined the current status of the hypoallergenic wheat developed for IgE-mediated wheat allergy typically showing a type of WDEIA.”

Comment#10: The last part of the introduction is marginally effective. Considering a review article, the introduction is quite summarized. It should explain more about the study. The authors may explain the pathophysiology of WA briefly.

Reply to the main comment #10. As we described in the Reply to the main comment #8, the role of this review is to serve an outline of development of hypoallergenic wheat for WDEIA. There have been many hypotheses and discussion on the role of co-factors inducing allergic reaction in WDEIA. We briefly summarized these in the Introduction section (line 60-64 in the revised manuscript) and added a reference no. 5 (in the revised version).

Comment#11: Lines no 65-67 discuss the theme of the study. The authors must give a little more information about such wheat and what efforts have been done so far and why the authors decided to write this manuscript.

Reply to the main comment #11. As we described in the Reply to the main comment #9, we added some description to the last part in the Introduction (line 79-82 in the revised manuscript) “However, hypoallergenic wheat to meet the patient’s needs has not been supplied. In order to analyze such approaches and to contribute to the further improvement, this study outlined the current status of the hypoallergenic wheat developed for IgE-mediated wheat allergy typically showing a type of WDEIA.”

Comment#12: The position of Table 1 is not correct. According to the ‘foods’ recommendations, Ideally, It should come somewhere after line no 83 (where the table has been mentioned for the very first time). The third column 'Reference' should display the name of the author and year with the citation. Also, note that ‘Table 1’ has been mentioned two times.

Reply to the main comment #12. According to your suggestion, Table 1 was moved after the line 83 (line 108 in the revised manuscript). Repetitive description of Table 1 was corrected. Thank you for your indication.

Comment#13: I am not sure if the author has conducted some lab work and showed the immunoblotting pictures in Figure 1 or if they have taken the gel pictures from some other work. If the authors are using gel pictures from any previously conducted study, the authors must declare this and take permission from the journal. 

Reply to the main comment #13. Figure 1 is our original study which would better explain the work by Kohno et al (reference no. 45 in the revised manuscript).

Comment#14: Authors must discuss the limitations of the study. 

 Reply to the main comment #14. We searched references with keyword of “hypoallergenic wheat” using PubMed, and 61 articles were extracted. Among them, 5 references are written in non-English languages. These references were omitted from the analysis in this study. This was described in the Conclusion section of the revised version (line 377-379).

Minor comments

Comment#1: Please discuss (in summary), why/how wheat triggers an immunological response. Line no. 33

Reply to the minor comment #1. This is a good question. Nobody can answer it correctly.

Comment#2: Why the authors used ‘were’ instead of ‘are’? If they are discussing any previous study, I think they should mention it. for example, according to so and so et al. Then the 'were' will be justified. Line no 88-90

Reply to the minor comment #2. Thank you for your comment. We corrected the “were” to “are”, in the revised version.

Comment#3: Which patients? I think again this line is a part of any previous study. if they use ‘were’ it means it is a previously conducted work. I suggest the authors if they are not using any previously conducted study better write the sentence in the present tense. Line no. 106

Reply to the minor comment #3. Thank you for your comment. We corrected the “were” to “are”, in the revised version.

Comment#4: If the mentioned work is a recently published study. the authors should use ‘has/have been’ instead of ‘were’. Line no 120-122

Reply to the minor comment #4. Thank you for your comment. We corrected the “were” to “have been”, in the revised version.

Comment#5: This heading can be more effective. Line no 123

Reply to the minor comment #5. We have rearranged the heading, according to your suggestion in the revised manuscript.

Comment#6: I suggest shifting reference#30 to line number 117. And if lines number 117-119 are part of the same study, cite the references 30-31 together. Also please shift reference#36 from line no 127 to line no. 136.

Reply to the minor comment #6. The author of reference no. 30 (no.31 in the revised manuscript) is different from those of reference no. 31 (no.32 in the revised manuscript). Reference No. 36 (no.37 in the revised manuscript) was shifted in the revised version. Thank you for your suggestion.

Comment#7: Please cite the reference at the end of the corresponding lines instead of adjacent to the author’s name.

Reply to the minor comment #7. We shifted the reference number at the end of the description through the manuscript in the revised version.

Comment#8: Lines number 44, and 162 need reference. 

Reply to the minor comment #8. We added the references after these descriptions.

Reviewer 3 Report

The manuscript was easy to ready and flowy. It covers a nice topic about the strategies of reducing wheat allergenicity to fit people having wheat allergy.

Main comments:

-the abstract needs restructuring to set the scope clear and the main outcomes of the review.

- the objective, in the end of introduction, need to be better introduced to show this work originality

-the sections are narrative, can you be more critical and enrich discussion?

-sum up table(s) of the different studies (and their main outcomes) to make hypoallergenic wheat would be helpful for a better read

Author Response

Reply to the comments

Comments from Reviewer 3

The manuscript was easy to ready and flowy. It covers a nice topic about the strategies of reducing wheat allergenicity to fit people having wheat allergy. 

Reply to the comment. Thank you for your favorable comment. We have studied your following comments carefully and made some corrections. We hope the revised manuscript will be suitable for publication.

Main comments: 

  • the abstract needs restructuring to set the scope clear and the main outcomes of the review.

Reply to the main comment 1. According to your comment, we reconstructed the abstract clearly to show the scope and outcome of the review.

  • the objective, in the end of introduction, need to be better introduced to show this work originality

Reply to the main comment 2. According to your comment, we changed the last sentence of the Introduction section “In order to analyze such approaches and contribute to the further improvement, this study outlined the current status of the hypoallergenic wheat developed for IgE-mediated wheat allergy typically showing a type of WDEIA.”

  • the sections are narrative, can you be more critical and enrich discussion?

Reply to the main comment 3. Thank you for your critical comment. We have studied your comment carefully, however we feel that it is not suitable for this review article to state large amount of discussion in each references. Therefore, the title was changed to “A narrative mini review on current status of hypoallergenic wheat developed for IgE-mediated wheat allergy, wheat-dependent exercise-induced anaphylaxis”.

  • sum up table(s) of the different studies (and their main outcomes) to make hypoallergenic wheat would be helpful for a better read

Reply to the main comment 4. Thank you for your suggestive comment. According to your comment, we added Table 2 summarizing the studies with their main outcomes and limitations. According to making this table, the reference number was rearranged. In addition, we added one article newly appeared describing a study using the mutant wheat cultivar as reference no. 49 in the revised manuscript.

Round 2

Reviewer 2 Report

I appreciate the authors for updating my knowledge about the difference between a short communication and a review article. I am also grateful for letting me know what should be the content in an original and review article.

With my little experience, I believe interpreting the result of previous work will be better than putting what the result was. Nevertheless, I agree with the authors and respect their responses.

The authors have done great work. I have some doubts only.

·         I understand the aim of the manuscript. the authors have suggested a number of articles to consult to know the definition, diagnosis, and pathophysiology of WDEIA. This was my concern,  for an article that comprehensively discusses the overall development of hypoallergenic wheat for WDEIA must mention the small points in the summary (4-6 lines). As a reader, I may not want to search for what is WDEIA in other articles.

·         I am not confirmed if an author has to take approval to show their previously published pictures from the respective journal. However, I believe the authors are way more informative than me.

·         Limitations of the study mean discussing some points that the authors could have done but they did not.

·         I understand that so far there is known pathophysiology of WDEIA is known but this could be written in a decent format to justify there is not enough information known so far ( e.g, WDEIA is caused by omega 5 gliadin component of wheat that triggers an IgE-mediated allergic response….)

Author Response

Comments and Suggestions for Authors by Reviewer 2

I appreciate the authors for updating my knowledge about the difference between a short communication and a review article. I am also grateful for letting me know what should be the content in an original and review article.

With my little experience, I believe interpreting the result of previous work will be better than putting what the result was. Nevertheless, I agree with the authors and respect their responses.

Reply to the comments and suggestions by Reviewer 2. Thank you for your comments and suggestions. We have studied your following comments carefully and made some corrections. We feel that your comments greatly helped improving the manuscript and hope the re-revised manuscript will be suitable for publication.

The authors have done great work. I have some doubts only.

[Comment 1]   I understand the aim of the manuscript. the authors have suggested a number of articles to consult to know the definition, diagnosis, and pathophysiology of WDEIA. This was my concern, for an article that comprehensively discusses the overall development of hypoallergenic wheat for WDEIA must mention the small points in the summary (4-6 lines). As a reader, I may not want to search for what is WDEIA in other articles. 

Reply to the comment 1. We added a brief explanation of WDEIA with relevant reference 6 in the Introduction section (line 60 - 69 in the re-revised manuscript).

[Comment 2]   I am not confirmed if an author has to take approval to show their previously published pictures from the respective journal. However, I believe the authors are way more informative than me.

Reply to the comment 2. The immunoblotting pictures presented in Figure 1 were newly made only for this review to support the results presented by Kohno et al (reference no. 46) and has not been published anywhere. Therefore, we need no approval to use them in this review.

[Comment 3]   Limitations of the study mean discussing some points that the authors could have done but they did not.

Reply to the comment 3. As you indicated, we have not presented details and of the methods which have been used for developing the hypoallergenic wheat. This description was added in the Discussion section (line 516-517 in the re-revised manuscript).

[comment 4]   I understand that so far there is known pathophysiology of WDEIA is known but this could be written in a decent format to justify there is not enough information known so far ( e.g, WDEIA is caused by omega 5 gliadin component of wheat that triggers an IgE-mediated allergic response….)

Reply to the comment. We added more information about WDEIA concerning the major responsible allergen and its usefulness in diagnosing WDEIA (line60 - 69 in the re-revised manuscript) as indicated in the reply to the comment 1.
